# Visualization of the Two-Phase Flow Behavior Involved in Enhanced Dense Phase Carbon Dioxide Pasteurization by Means of High-Speed Imaging

Ratka Hoferick [1], Holger Schönherr [1] and Stéphan Barbe [2,*]

1 Department of Chemistry and Biology, Physical Chemistry I & Research Center of Micro and Nanochemistry and (Bio)Technology (Cμ), University of Siegen, Adolf-Reichwein-Str. 2, 57076 Siegen, Germany; ratka.hoferick@gmail.com (R.H.); schoenherr@chemie.uni-siegen.de (H.S.)
2 Technische Hochschule Köln, Faculty of Applied Natural Sciences, Campus Leverkusen, Campusplatz 1 Leverkusen, 51379 Leverkusen, Germany
* Correspondence: stephan.barbe@th-koeln.de

**Abstract:** This research explores the two-phase flow behavior involved in enhanced dense phase carbon dioxide inactivation of *E. coli* DH5α, which has been shown to possess a high microbial reduction efficiency of up to $3.7 \pm 0.4$ log. We present an experiment in which the liquid sample was pressurized with liquid carbon dioxide to 8.2 MPa and, after saturation, was forced to flow through a mini tube. An experimental setup was developed to visualize the flow patterns (plug, slug and churn flows) occurring in the mini tube by means of high-speed imaging. The values of the wall shear stress were estimated within the mini tube with the help of the gas slug velocities (8–9 m/s) and were compared with threshold shear stress values reported for the disruption of fresh *E. coli* cells. The results suggest that the preliminary pressurization phase may cause a substantial destabilization of the cell wall of *E. coli* DH5α.

**Keywords:** high-speed imaging; multiphase flow; experimental fluid mechanics; dense phase carbon dioxide; microbial inactivation; mechanical effects





## 1. Introduction

Microbial inactivation induced by Dense Phase Carbon Dioxide (DPCD) is a well-known non-thermal pasteurization technique that allows extending the shelf life of heat-sensitive liquid food products (e.g., juices) without severely affecting their organoleptic properties (fresh-like products) [1]. During conventional DPCD, satisfying inactivation results are usually achieved by pressurizing liquids with supercritical carbon dioxide in a pressure vessel for 5 to 60 min. Typically, temperatures between 35 and 50 °C and pressures between 10 and 50 MPa are applied during DPCD [2]. The necessity to work at such high pressures makes both the technical realization and the scaling up of this process quite challenging. Operating DPCD at lower pressure usually requires synergetic effects from the additional application of ultrasound or stirring or the use of additional chemical reactants like nisin in order to achieve acceptable microbial inactivation (reduction >3 log) [3,4].

The inactivation mechanisms involved in DPCD are not yet fully understood [5–7]. Amongst other effects, the destabilization of the bacterial cell wall caused by the penetration of carbon dioxide has been identified as one of the major inactivation phenomena [8]. Indeed, the diffusion of carbon dioxide through the lipid bilayer of biological membranes is often accompanied by critical mechanical and structural changes (expansion and swelling, disturbance of membrane fluidity, disordering of the lipid bilayer) that usually lead to severe membrane dysfunctions [9–11]. Acting as a protective barrier, the biological membrane plays a crucial role in the survival of bacteria, and the loss of the membrane's mechanical stability may dramatically reduce bacterial resilience to physical constraints, such as friction, stretching and shearing [11].

In this context, it is well known that bacterial cells can be disrupted by shear forces [8]. In 2001, Lange et al. [9] developed an experimental setup to determine the resistance of the cell walls of *E. coli* to disruption. For this purpose, freshly harvested *E. coli* cells (strain 188) were exposed to shear stress (in yeast extract at 30 °C) that was initiated by a laminar flow through a narrow capillary (inner diameter 0.096 to 0.183 mm, length 50 to 400 mm). The authors reported the disruption of *E. coli* cells at shear stress values above 1.8 kN/m$^2$ and estimated a threshold of about 1.3 kN/m$^2$.

We have recently demonstrated the possibility to dramatically enhance DPCD through mechanical effects to achieve satisfactory inactivation results under milder conditions (pressure < 8.5 MPa) [2,12]. During this modified process, which is referred to as Multi-Phase-Pressure-Drop microbial (MPPD) inactivation, the liquid is first pressurized in a high-pressure vessel (8.2 MPa) with carbon dioxide at a temperature of 30 °C. After a sudden depressurization, the liquid saturated with carbon dioxide is forced to flow through a mini tube (inner diameter 0.57 mm, length 230 mm) and is subjected to an acceleration leading to high values of flow velocity. The end of the mini tube is connected to a low-pressure vessel (at atmospheric pressure) in which the treated liquid is finally collected.

In the framework of an extensive experimental investigation, we also showed that the effect of this post-treatment has clear statistical significance [2]. Compared to the segment of conventional DPCD, the inactivation efficiency for *E. coli* DH5$\alpha$ rose from a 0.2 $\pm$ 0.2 log to a 3.7 $\pm$ 0.4 log reduction for MPPD. Consequently, this process was recognized as a very promising pathway to realize a new cost-efficient DPCD technology. Furthermore, the inactivation effect of MPPD was confirmed with fluorescence microscopy combined with a live/dead staining assay. Images of the bacterial cells obtained by Field Emission Scanning Electron Microscopy (FESEM) and Atomic Force Microscopy (AFM) clearly revealed that the disruption of *E. coli* cells occurred during the MPPD trials, and cell debris was detected surrounding the bacteria [2]. With pressures up to 77 bar, the pressure drop associated with the MPPD experiments is high. Consequently, sudden degassing, high flow velocities and high shear stresses are expected to occur during the discharge of the liquid from the pressure vessel via the mini tube to the low-pressure vessel. We hypothesized that high shear stress in the mini tube is the main cause for the extensive cell disruption observed after MPPD treatment [2], but efforts are needed to confirm this hypothesis. Substantial progress was recently achieved in describing, analyzing and modeling complex flow processes [13–15].

The main objective of the present contribution is to get in situ insight into the complex transient and turbulent two-phase flow behavior that occurs in the mini tube during MPPD trials. For this purpose, an experimental setup was developed in order to explore the corresponding flow patterns by means of high-speed imaging technology and estimate the shear stress at the wall of the mini tube, where maximum values are expected to arise during MPPD. Furthermore, this work aims to analyze the flow processes in much more detail and in particular to retrace the sequence of events that leads to the high inactivation of *E. coli* during MPPD in order to provide a better understanding of the inactivation mechanisms involved in this process.

## 2. Materials and Methods

### 2.1. High-Speed Imaging

In order to visualize the two-phase flow occurring in the mini tube during MPPD experiments, the setup used by Hoferick et al. [2,12] was modified (Figure 1). The following parameters differ from the original layout because they had to be adjusted for the purpose of video imaging. The stainless-steel mini tube (length 230 mm, inner diameter 0.57 mm and cross section 0.255 mm$^2$) was substituted by a transparent fluorinated ethylene propylene (FEP) mini tube (length 230 mm, inner diameter 0.50 mm and cross section 0.196 mm$^2$) that was immersed in a water bath to optimize the refractive index contrast. Furthermore, bacteria-free LB medium (Luria/Miller Carl Roth GmbH, Karlsruhe, Germany) was used. Video images were recorded by a Photron FASTCAM Mini AX200 type 900K-M-8GB (27,000 fps). The light source was a high intensity LED fiber optic illuminator

(SugarCUBE™, USHIO, Vergennes, VT, USA). A thin sheet of paper was placed between the illuminator and the mini tube to spread the light (Figure S1a,b in Supporting Information).

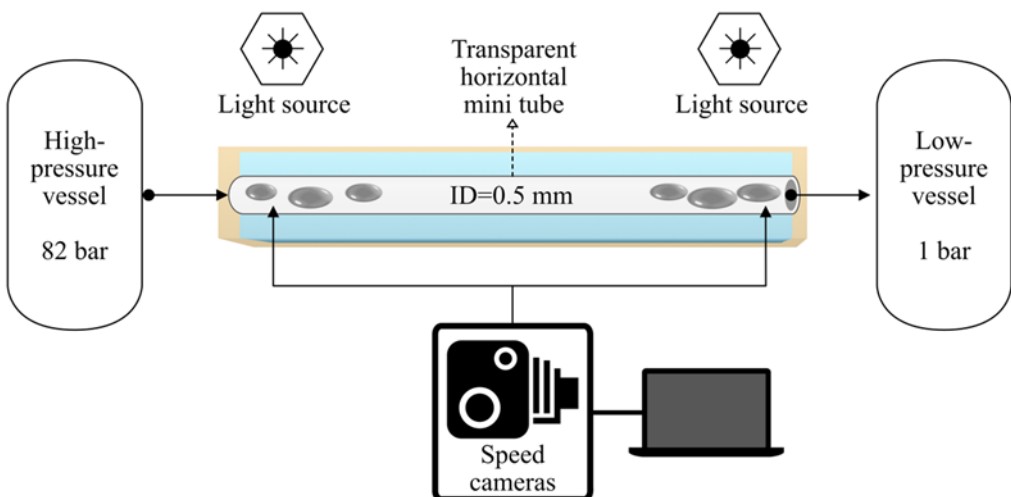

**Figure 1.** Scheme of the experimental setup used in this work.

Prior to each MPPD trial, the high-pressure vessel (Parr Instrument GmbH, Paar Instruments, Moline, IL, USA) was filled with 50 mL of LB medium and was pressurized with liquid carbon dioxide (99.995% (*v/v*) (Messer Industriegase GmbH, Siegen, Germany) at 30 °C. Subsequently, the inactivation pressure (p1) in the high-pressure vessel was set to 8.2 MPa and the recording function of the camera was activated. The experiment started with the rapid opening of the 90° needle valve located at the top of the high-pressure vessel. Video recordings and video images were processed with the software Photron FASTCAM Viewer Application (Ver.3).

For the verification of the high-speed-imaging-technology (HSIT): Figure 2a shows only gas (air) in the mini tube; Figure 2b shows the conditions of combined liquid and gaseous $CO_2$ under pressure used for all video imaging experiments. At 8.2 MPa and 30 °C, carbon dioxide is in its liquid state and evaporates by entering the mini tube due to the pressure drop. The liquid phase includes water and/or liquid $CO_2$, which can be discerned by the bright-colored parts of the images. The dark-colored parts of the images represent the gaseous phase or, to be more precise, $CO_2$ gas bubbles.

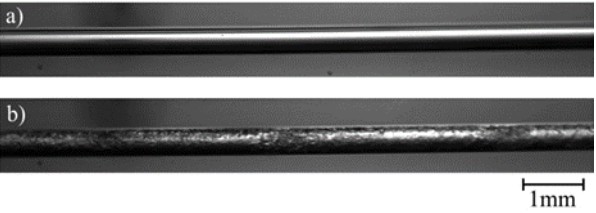

**Figure 2.** Photographs of the transparent mini tube filled with (**a**) air and (**b**) carbon dioxide.

### 2.2. Estimation of Wall Shear Stress during Two-Phase Flow in the Mini Tube

As mentioned above, the main characteristics of MPPD are a transient and complex turbulent two-phase flow occurring in the mini tube. This regime is quite different from the one used in conventional gas–liquid two-phase flow investigations. The latter are characterized by steady flow conditions and the injection of gas and liquid through a mini tube at constant flow rates. Gas–liquid flow in mini tubes often results in the formation of slug flow patterns, in which a cylindrical gaseous slug is surrounded by a liquid film [16,17]. The two following approaches were proposed for the estimation of the corresponding wall shear stress.

### 2.2.1. Steady-State Slug Flow

For this approach, the steady-state slug flow and the concept of unit cell was applied. The flow is considered as a sequence of cells that are periodic both in space and time [18].

The wall shear stress near gas slugs is expressed by Laborie et al. [18] as

$$\tau_{LGas} = \frac{\rho_L \, f_{Gas} \, U^2_{LGas}}{2} \tag{1}$$

where $\rho_L$ is the density of the liquid (water), $f_{Gas}$ denotes the friction factor near the gas slug and $U_{LGas}$ the liquid velocity near the gas slug [11]. $f_{Gas}$ and $U_{LGas}$ are calculated as:

$$f_{Gas} = \frac{16}{Re_{Gas}} \tag{2}$$

where $Re_{Gas}$ is the Reynolds number characterizing the liquid flow near the gas slugs, calculated as:

$$Re_{Gas} = \frac{U_{LGas} \, D_{Gas} \, \rho_L}{\mu_L} \tag{3}$$

where $D_{Gas}$ is the hydraulic diameter near the gas slug and $\mu_L$ the viscosity of the liquid (water). $D_{Gas}$ is expressed as:

$$D_{Gas} = 4 \, e \tag{4}$$

where $e$ is liquid film thickness. While it is not possible to measure $e$ based on video imaging, it is written as:

$$e = \frac{r}{2} \sqrt{\frac{V_s \, \mu_L}{\sigma_L}} \tag{5}$$

where $r$ is the radius of the mini tube, $V_s$ is the gas slug velocity, which can be calculated based on video imaging as the distance traveled in time and $\sigma_L$ is surface tension of the liquid (water).

$$U_{LGas} = \frac{-\Phi_L}{R_{LGas}} + V_s \tag{6}$$

where $\Phi_L$ is liquid flux between a gas slug and a liquid slug and $R_{LGas}$ the liquid holdup in the gas slugs.

$$\Phi_L = R_L \, V_S - U_{Liq} \tag{7}$$

where $R_L$ is liquid holdup and $U_{Liq}$ is liquid superficial velocity for a two-phase flow.

$$R_L = 1 - \frac{V^G}{V} \tag{8}$$

where $V^G$ is a volume of the gas phase and $V$ the total volume (gas + liquid) based on video images. For the calculation of $R_L$ we assumed that the gas and liquid slug possess a cylindric form.

$$U_{Liq} = \frac{Q_{Liq}}{\pi \, r^2} \tag{9}$$

where $Q_{Liq}$ is the liquid flow rate that is not possible to be estimated from this experimental video imaging as we do not have a specific liquid motive to follow up. Instead of $U_{Liq}$, the superficial gas velocity for a two-phase flow $U_{Gas}$ can be estimated.

$$U_{Gas} = \frac{Q_{Gas}}{\pi \, r^2} \tag{10}$$

where $Q_{Gas}$ is the gas flow rate. For the calculation of $Q_{Gas}$ as a volume of the gas slug in time based on video imaging, we assumed that the gas slug possesses a cylindric form. With the information about $U_{Gas}$ it is possible to define the range of $U_{Liq}$ using the diagram

for the slug flow area [16]. Liquid holdup in the gas slugs $R_{LGas}$ was calculated with Equation (11) as [17]:

$$R_{LGas} = \frac{2\,e}{r} \tag{11}$$

where Equation (5) for liquid film thickness was applied and the mini tube radius is known. The wall shear stress near liquid slugs is expressed as:

$$\tau_{LLiq} = \frac{\rho_L\,f_{Liq}\,U^2_{LLiq}}{2} \tag{12}$$

where $f_{Liq}$ is the friction factor near the liquid slug and $U_{LLiq}$ the liquid velocity near the liquid slug [11].

$$f_{Liq} = \frac{16}{Re_{Liq}} \tag{13}$$

To calculate the Reynolds number characterizing the liquid flow in the liquid slugs, $Re_{Liq}$, information about $U_{LLiq}$ and $D_{Liq}$ the hydraulic diameter near the liquid slug is needed:

$$Re_{Liq} = \frac{U_{LLiq}\,D_{Liq}\,\rho_L}{\mu_L} \tag{14}$$

$$U_{LLiq} = (1 - R_L)\,V_s + U_{Liq} \tag{15}$$

$$D_{Liq} = 2\,r \tag{16}$$

### 2.2.2. Single Flow Approach

In this case, the two-phase flow was simplified as a steady-state single-phase flow and the average liquid flow velocity $_L$ was set equal to the bubble velocity. The corresponding values were derived from high-speed imaging. Subsequently, the wall shear stress $\tau_W$ was calculated as:

$$\tau_W = \frac{f}{8} \cdot \left(\rho_L \cdot v_L^2\right) \tag{17}$$

where the corresponding friction factor $f$ was calculated by using the Hermann correlation [19]:

$$f = 0.0054 + \frac{0.3964}{Re_{Liq}^{0.3}} \tag{18}$$

### 3. Results and Discussion

The investigation of the flow behavior inside the mini tube was carried out by means of high-speed imaging using a designated setup. Selected videos and further details of the experimental setup can be found as published supporting information. Figures 3 and 4 show the two-phase patterns visualized at the beginning and at the end of the mini tube, respectively (full videos are available in supporting material SV1—beginning of the mini tube and SV2—end of the mini tube). A first striking observation is the absence of a discernible transition from a single to a two-phase flow via e.g., bubbly flow. The flow pattern observed at the beginning of the mini tube consists of long Taylor bubbles in a gas–liquid slug flow. This suggests that considerable transient degassing already occurred in the peripheral devices in front of the mini tube leading to an increase of the void fraction and the observed slug flow. Furthermore, the observed flow is quite homogeneous and stable in time and space, which supports our approach for the estimation of wall shear stress values based on steady-state flow approximations. Based on the video imaging, we were able to estimate superficial velocities of the corresponding slugs (Table 1).

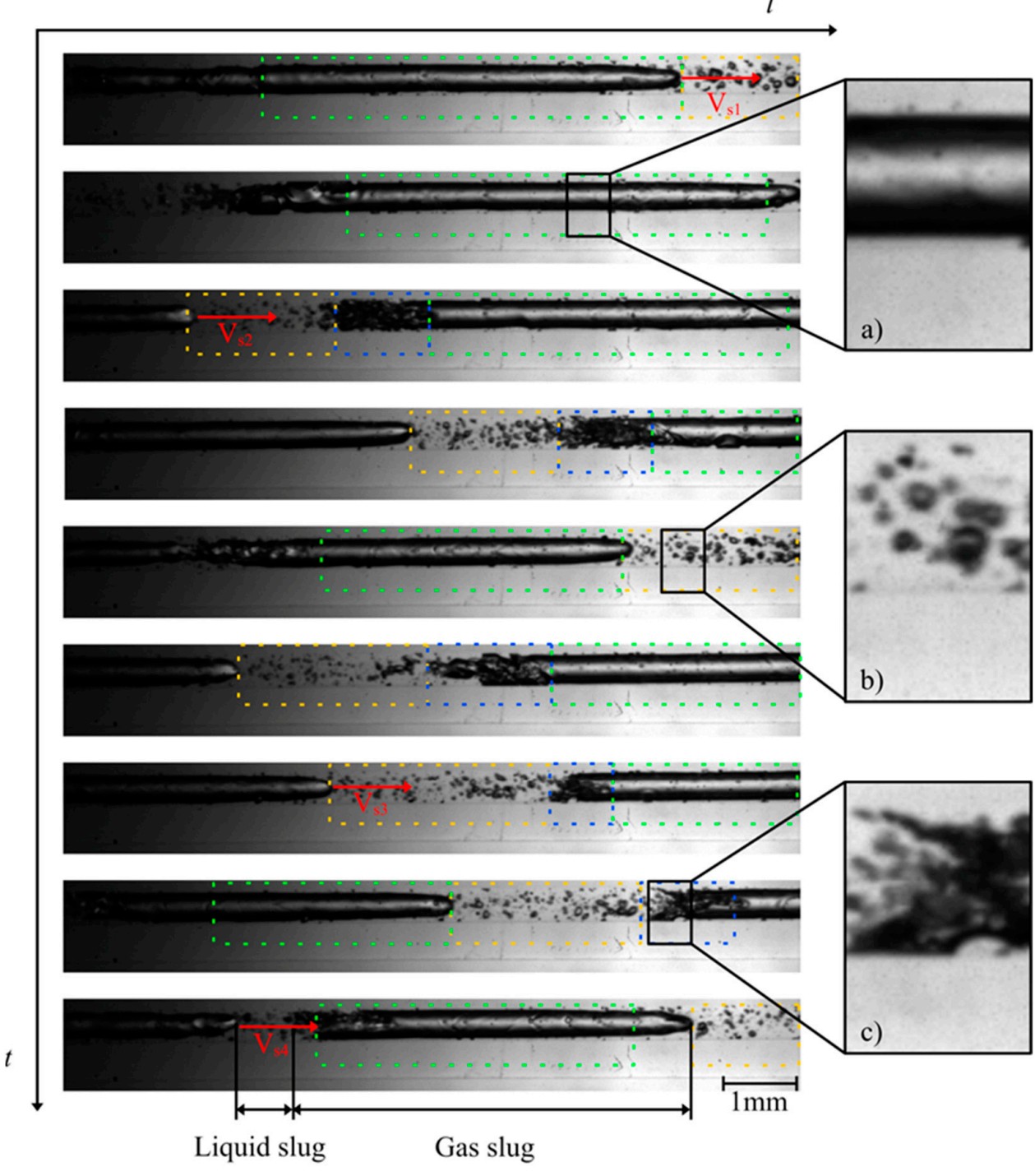

**Figure 3.** Photographs of the flow patterns observed at the beginning of the mini tube during MPPD in 3 µs (at 82 bar starting inactivation pressure): (**a**) gas plug in the transition to a gas slug (green dashes); (**b**) liquid plug (yellow dashes) and (**c**) churn flow (blue dashes). Full video is available in supporting material SV1.

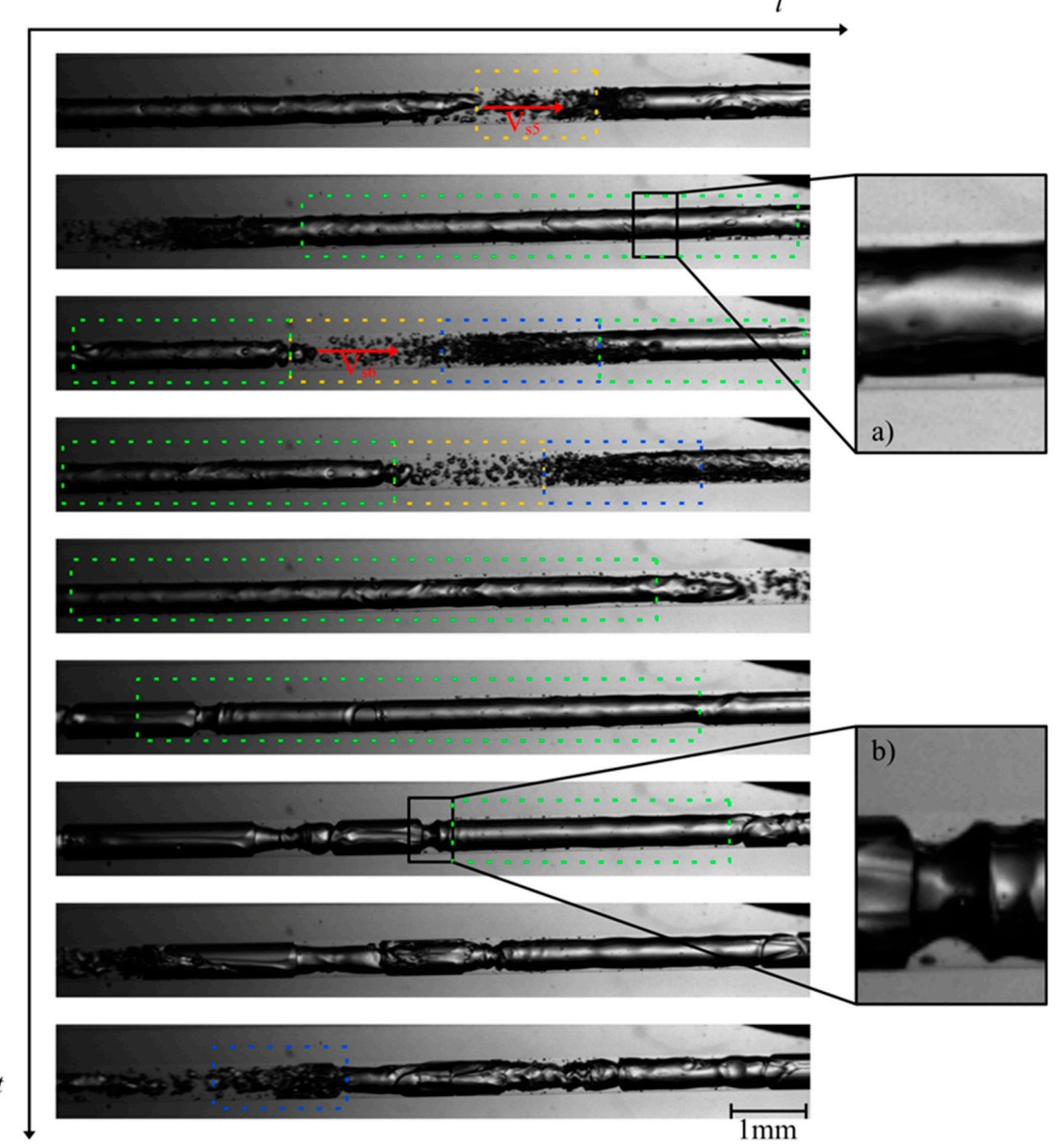

**Figure 4.** Photographs of the flow patterns at the end of the mini tube during MPPD in 6 μs (at 82 bar starting inactivation pressure): slug flow example (green dashes), liquid slugs example (yellow dashes) and churn flow example (blue dashes); (**a**) thin liquid layer on the wall of the mini tube and (**b**) transition to churn flow. Full video is available in supporting material SV2.

**Table 1.** Average slug velocities at the beginning and at the end of the mini tube for Δp = 7.7 MPa.

| Average Gas Slug Velocity at the Beginning of the Mini Tube ($\pm 0.05$ m/s) | Average Gas Slug Velocity at the End of the Mini Tube ($\pm 0.05$ m/s) |
|:---:|:---:|
| $V_{S1,b} = 8.1$ m/s | $V_{S1,e} = 9.1$ m/s |
| $V_{S2,b} = 9.1$ m/s | $V_{S2,e} = 9.0$ m/s |
| $V_{S3,b} = 9.0$ m/s | $V_{S3,e} = 9.0$ m/s |

In this work, we estimated the flow velocity $V_S$ at different positions e.g., 1,2 and 3 leading to the indexes $V_{S1}$, $V_{S2}$ and $V_{S3}$, respectively. Ranging from 8.0 to 9.1 m/s, the gas slug moved at high velocity within the mini tube. At the end of the large Taylor bubbles, a churn flow (blue dashes in Figures 3 and 4) can be observed, characterized by many small bubbles which can interact with each other and form new bubbles. A clear transition to

churn flow finally appears at the end of the mini tube (Figure 4). We applied the steady-state slug flow approach by using the average slug velocity from Table 1 in Equations (1)–(16) and calculated wall the shear stress near gas slug $\tau_{LGas}$ = 0.3–0.4 kN/m$^2$ and the wall shear stress in liquid slug $\tau_{LLiq}$ = 0.1–0.2 kN/m$^2$. The single flow approach led to comparable wall shear stress values $\tau_W$ = 0.3–0.4 kN/m$^2$. Consequently, the wall shear stress during MPPD values is much lower than the threshold shear stress value determined by Lange et al. [9] for the disruption of fresh *E. coli* cells (1.3 kN/m²). One conceivable explanation for this observation is that, due to the pressurization step and the diffusion of $CO_2$ through the lipid bilayer, the cell wall of *E. coli* is fragilized and becomes more sensitive to shear stress during MPPD inactivation trials.

### 4. Conclusions

This work provides crucial in situ insight into two-phase flow behavior that occurs in the mini tube during the MPPD process. Due to the ongoing degassing and increase of the void fraction, a transition to churn flow was observed at the end of the mini tube. We updated our hypothesis concerning the inactivation phenomena involved in MPPD. In this context, the present contribution reveals the potential key role that the pressurization phase may have on the alteration of the membrane of *E. coli* cells. According to this hypothesis, the bacteria would become less resilient to shear stress and could be disrupted by applying lower shear stress values.

Further efforts are still needed to confirm this hypothesis and gain better understanding of the inactivation mechanisms involved in this promising non-thermal pasteurization technique for liquid foods. In this regard, our research activities will focus on the detailed investigation and optimization of the alteration of the membrane of *E. coli* cells through $CO_2$ pretreatment. The determination of threshold shear stress efficiencies for *E. coli* cells with altered membranes may be decisive. Furthermore, the efficiency of MPPD in the horizontal mini tube may depend on several factors, such as mini tube diameter, pressure drop, wettability, surface roughness, density, viscosity of the liquid part and surface tension. During our recent MPPD experiments, we realized that length of the mini tube may severely affect the overall microbial inactivation and we identify it as a key parameter during upscaling strategies. It is also our plan that some of future video imaging focuses on flow patterns and velocities by varying the pressure drop.

### 5. Patents

The presented method has been submitted for patent registration under the number DE 10 2017 011 752.7.

**Supplementary Materials:** The following supporting information can be downloaded at: https://www.mdpi.com/article/10.3390/fluids9010010/s1, Figure S1, Videos S1 and S2: High speed videos of the two-phase flow behavior observed in the mini tube during enhanced DPCD.

**Author Contributions:** Conceptualization, R.H., H.S. and S.B.; methodology, R.H. and S.B.; software, R.H.; formal analysis, R.H., H.S. and S.B.; investigation, R.H.; resources, H.S. and S.B.; data curation, R.H., H.S. and S.B.; writing—original draft preparation, R.H.; writing—review and editing, R.H., H.S. and S.B.; visualization, R.H.; supervision, H.S. and S.B.; project administration, S.B. All authors have read and agreed to the published version of the manuscript.

**Funding:** This research received no external funding.

**Data Availability Statement:** Selected videos and further details of the experimental setup can be found as Supporting Information.

**Acknowledgments:** The authors gratefully acknowledge the Messer Group GmbH and Messer Industriegase GmbH for cooperation and thank Guido Simon, Davor Spoljaric, Frank Gockel, Claus-Dieter Ohl, Katja Guttmann, Fabian Reuter, Wolfgang Jantoss, Patricia Pfeiffer, Kristin Hecht, Steve Hoferick, Mareike Müller and Tobias Wolf for inspiring discussions and their excellent technical support.

**Conflicts of Interest:** The presented method has been submitted for patent registration under the number DE 10 2017 011 752.7.

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
