# Peer review of "Visualization of the Two-Phase Flow Behavior Involved in Enhanced Dense Phase Carbon Dioxide Pasteurization by Means of High-Speed Imaging"

_fluids, doi:10.3390/fluids9010010_

Round 1

Reviewer 1 Report

Comments and Suggestions for Authors

The authors propose an experiment called MPPD that achieves a very high microbial fire suppression efficiency under moderate pressure conditions. An experimental setup was developed to visualize the flow patterns (plug, slug and churn flows) occurring in a mini tube by means of high-speed imaging. The values of the wall shear stress were estimated within the mini tube with the help of the gas slug velocities (8-9 m/s) and compared with threshold shear stress values reported for the disruption of fresh E. coli cells. The conclusion was drawn that the preliminary pressurization phase may cause a substantial destabilization of the cell wall of E. Coli DH5α. However, some revisions are required before the manuscript is accepted for publication. The authors should address the following comments:

1. The manuscript mentions that in the MPPD process, there is an ongoing degassing and increase of the void fraction, which inevitably leads to a flow velocity at the outlet of the mini tube being greater than the flow velocity at the inlet of the mini tube. However, from Table 1 in the manuscript, it can be seen that there has not been a significant change in the flow rate. Please explain this conflict.

2. The manuscript mentions that in MPPD experiments, length of the mini tube may severely affect the overall microbial inactivation. At the same time, the manuscript also mentioned that a transition to churn flow was observed at the outlet of the mini tube. Therefore, the increase in microbial inactivation rate in MPPD experiments is likely due to cell wall damage caused by shear stress exceeding the threshold in the churn flow. The manuscript does not provide direct evidence that the preliminary pressurization phase causes a substantial destabilization of the cell wall of E. Coli DH5α, nor does it calculate the maximum shear stress in the churn flow. Therefore, the results of the current high-speed imaging experiments are not sufficient to show that a certain hypothesis is correct. Please add relevant experiments.

3. Only 16 reference is not enough. Some related and recent work can be added, e.g. An integrated model with stable numerical methods for fractured underground gas storage. Journal of Cleaner Production, 2023, 393, 136268.  A novel complex network-based deep learning method for characterizing gas–liquid two-phase flow. Pet. Sci. (2020). https://doi.org/10.1007/s12182-020-00493-3.  An efficient multigrid-DEIM semi-reduced-order model for simulation of single-phase compressible flow in porous media. Pet. Sci. 18, 923–938 (2021). https://doi.org/10.1007/s12182-020-00509-y.

4. The use of units in section 2.1 of the manuscript is not standardized, e.g. the 2 in the unit of area mm2 should be superscripted.

5. The labels in Figures 3 and 4 are so blurred that it is impossible to distinguish between VS2, VS3, VS4, VS5, etc. Please re-upload the clearly labeled images.

6. Please explain the physical meaning of VS1, VS2, and VS3 and why the velocity zone should be divided into VS1, VS2, and VS3.

Author Response

Dear reviewer,

Thank you very much for your helpful feedback and comments. We dedicated considerable attention to address each point of your review.

  1. The manuscript mentions that in the MPPD process, there is an ongoing degassing and increase of the void fraction, which inevitably leads to a flow velocity at the outlet of the mini tube being greater than the flow velocity at the inlet of the mini tube. However, from Table 1 in the manuscript, it can be seen that there has not been a significant change in the flow rate. Please explain this conflict.

RESPONSE: As mentioned in the first paragraph of section 3: “A first striking observation is the absence of a discernible transition from a single to a two-phase flow via e.g. bubbly flow. The flow pattern observed at the beginning of the mini tube consists of long Taylor bubbles in a gas–liquid slug flow. This suggests that considerable transient degassing already occurred in the peripheral devices in front of the mini tube leading to an increase of the void fraction and the observed slug flow.”

  1. The manuscript mentions that in MPPD experiments, length of the mini tube may severely affect the overall microbial inactivation. At the same time, the manuscript also mentioned that a transition to churn flow was observed at the outlet of the mini tube. Therefore, the increase in microbial inactivation rate in MPPD experiments is likely due to cell wall damage caused by shear stress exceeding the threshold in the churn flow. The manuscript does not provide direct evidence that the preliminary pressurization phase causes a substantial destabilization of the cell wall of E. Coli DH5α, nor does it calculate the maximum shear stress in the churn flow. Therefore, the results of the current high-speed imaging experiments are not sufficient to show that a certain hypothesis is correct. Please add relevant experiments.

RESPONSE: In this short communication, for the first time, we examine the flow involved in this very promising non-thermal pasteurization technique. Out of this work and under consideration of the current theoretical background of this research (as summarized in the introduction), we formulate the following new hypothesis “the preliminary pressurization phase causes a substantial destabilization of the cell wall of E. Coli DH5α”. It is true that we do not provide evidence for this new hypothesis and we do not claim to do this. In section 3, we rephrased the following sentence:

“This observation suggests that, due to the pressurization step and the diffusion of CO2 through the lipid bilayer, the cell wall of E. Coli is fragilized and becomes more sensi-tive to shear stress during MPPD inactivation trials.”

In “One conceivable explanation for this observation is that, due to the pressurization step and the diffusion of CO2 through the lipid bilayer, the cell wall of E. Coli is fragilized and becomes more sensi-tive to shear stress during MPPD inactivation trials.”

The necessity to dedicate efforts in further experiments was made clear in our conclusion (section 4) as we wrote:

 “Further efforts are still needed to confirm this hypothesis and gain better understanding of the inactivation mechanisms involved in this promising non-thermal pasteurization technique for liquid foods. In this regard, our research activities will focus on the detailed investigation and optimization of the alteration of the membrane of E. Coli cells through CO2 pretreatment. The determination of threshold shear stress efficiencies for E. coli cells with altered membranes may be decisive.”

We also rephrased our statement as follows:

“We updated our hypothesis concerning the inactivation phenomena involved in MPPD. In this context, the present contribution reveals the potential key role that the pressurization phase may have on the alteration of the membrane of E. Coli cells. According to this hypothesis, the bacteria would become less resilient to shear stress and can be disrupted by applying lower shear stress values.”

  1. Only 16 reference is not enough. Some related and recent work can be added, e.g. An integrated model with stable numerical methods for fractured underground gas storage. Journal of Cleaner Production, 2023, 393, 136268. A novel complex network-based deep learning method for characterizing gas–liquid two-phase flow. Pet. Sci. (2020). https://doi.org/10.1007/s12182-020-00493-3. An efficient multigrid-DEIM semi-reduced-order model for simulation of single-phase compressible flow in porous media. Pet. Sci. 18, 923–938 (2021). https://doi.org/10.1007/s12182-020-00509-y.

RESPONSE: Thank you, we cited the recommended papers and added them to the reference list.

  1. The use of units in section 2.1 of the manuscript is not standardized, e.g. the 2 in the unit of area mm2 should be superscripted.

RESPONSE: Thank you, we checked the manuscript and corrected accordingly.

  1. The labels in Figures 3 and 4 are so blurred that it is impossible to distinguish between VS2, VS3, VS4, VS5, etc. Please re-upload the clearly labeled images.

RESPONSE: We have images in very good resolution in our word file with clear and readable labels. I guess resolution got lost during formatting to pdf.

  1. Please explain the physical meaning of VS1, VS2, and VS3 and why the velocity zone should be divided into VS1, VS2, and VS3.

RESPONSE: In this work, we estimated the flow velocity VS at different positions e.g. 1,2 and 3 leading to VS1, VS2 and VS3, respectively. We added this explanation in the manuscript.

Reviewer 2 Report

Comments and Suggestions for Authors

The Authors presented this communication in the proper way. It has all the parts that are needed for such a manuscript.

Author Response

Thank you for your feedback.

Reviewer 3 Report

Comments and Suggestions for Authors

The manuscript explores the two-phase flow behavior involved in enhanced dense phase carbon dioxide inactivation of E. Coli DH5α. In my opinion, the manuscript is publishable in the Fluids in its current form with minor explanations.

The manuscript is dealing with destabilization of the cell wall of E. Coli DH5α by the enhanced dense phase carbon dioxide inactivation. The use of supercritical fluid carbon dioxide has been discussed to control the activity of several bacteria strains by the shear stress and the pressure release. My question is how the operating conditions have been determined? is the applied pressure the optimal? The entire behaviour is probably closely related to the density and surface tension of the media, as it is concluded by the authors, that would give an additional value to the present research.

Comments on the Quality of English Language

Minor improvements required.

Author Response

Dear reviewer,

thank you very much for your feedback.

The operating condition and optimal pressure were derived from a recent paper that published in the Journal of Supercritical Fluids (Reference 2 in this manuscript).

Hoferick, A. Ntovas, Q. Alhusaini, M. Müller, S. Barbe, H. Schönherr, Enhanced microbial inactivation by carbon dioxide through mechanical effects, J Supercrit Fluids. 175 (2021) 105273. https://doi.org/10.1016/j.supflu.2021.105273

 In this work we carried out a systematic study of the impact of the operating conditions on the inactivation efficiency for E. Coli DH5α.

Round 2

Reviewer 1 Report

Comments and Suggestions for Authors

All the review comments have been addressed properly. This manuscript can be considered to be accepted.